Testing animal-assisted cleaning prior to transplantation in coral reef restoration

Frias-Torres Sarah 1 2 sfriastorres@gmail.com
van de Geer Casper 1 3
1 Nature Seychelles , Amitie, Praslin , Republic of Seychelles
2 Smithsonian Marine Station , Fort Pierce, FL , USA
3 Local Ocean Trust , Watamu , Kenya
Costello Mark
Electronic publication date: 2015 Sep 29
Publication date: 2015
Volume: 3
Electronic Location ID: e1287
Received 2015 Jul 7; Accepted 2015 Sep 11
Copyright: © 2015 Frias-Torres & van de Geer
Copyright year: 2015
Copyright holder: Frias-Torres & van de Geer
License: This is an open access article distributed under the terms of the Creative Commons Attribution License, which permits unrestricted use, distribution, reproduction and adaptation in any medium and for any purpose provided that it is properly attributed. For attribution, the original author(s), title, publication source (PeerJ) and either DOI or URL of the article must be cited.
License URL: https://creativecommons.org/licenses/by/4.0/

Keywords: Barnacle, Biofouling, Coral gardening, Indian ocean, Nursery, Seychelles, Transplantation, Biomimicry, Cleaning station

Funding: United States Agency for International Development (USAID) Reef Rescuers Project 674-A-00-10-00123-00 Funding to Nature Seychelles was received through the United States Agency for International Development (USAID) Reef Rescuers Project 674-A-00-10-00123-00. The funders had no role in study design, data collection and analysis, decision to publish, or preparation of the manuscript.

==============================
Rearing coral fragments in nurseries and subsequent transplantation onto a degraded reef is a common approach for coral reef restoration. However, if barnacles and other biofouling organisms are not removed prior to transplantation, fish will dislodge newly cemented corals when feeding on biofouling organisms. This behavior can lead to an increase in diver time due to the need to reattach the corals. Thus, cleaning nurseries to remove biofouling organisms such as algae and invertebrates is necessary prior to transplantation, and this cleaning constitutes a significant time investment in a restoration project. We tested a novel biomimicry technique of animal-assisted cleaning on nursery corals prior to transplantation at a coral reef restoration site in Seychelles, Indian Ocean. To determine whether animal-assisted cleaning was possible, preliminary visual underwater surveys were performed to quantify the fish community at the study site. Then, cleaning stations consisting of nursery ropes carrying corals and biofouling organisms, set at 0.3 m, 2 m, 4 m, 6 m and 8 m from the seabed, were placed at both the transplantation (treatment) site and the nursery (control) site. Remote GoPro video cameras recorded fish feeding at the nursery ropes without human disturbance. A reef fish assemblage of 32 species from 4 trophic levels (18.8% herbivores, 18.8% omnivores, 59.3% secondary consumers and 3.1% carnivores) consumed 95% of the barnacles on the coral nursery ropes placed 0.3 m above the seabed. Using this cleaning station, we reduced coral dislodgement from 16% to zero. This cleaning station technique could be included as a step prior to coral transplantation worldwide on the basis of location-specific fish assemblages and during the early nursery phase of sexually produced juvenile corals.

Introduction

Active coral reef restoration is increasingly being seen as a new tool for conservation biology (Precht, 2006) as coral reefs continue to decline worldwide (Hoegh-Guldberg, 2004). One of the several available coral reef restoration methods involves “coral gardening” in a two-step process. First, coral fragments are raised in underwater nurseries. Second, after reaching a target size, the nursery corals are harvested and transplanted onto degraded reef areas (Rinkevich, 2006).

The cleaning of algae and sessile invertebrates (sponges, hydroids, barnacles, mollusks, and tunicates) in nurseries is essential to avoid the space competition with corals, which leads to shading and coral death. Nursery cleaning consumes a significant portion of the time invested in restoration projects (Precht, 2006). Prior to transplantation onto reefs, the nursery corals require additional cleaning. Grazers and invertivores at the nursery site and adjacent natural reefs can potentially consume biofouling organisms, which reduces the need for human maintenance of the nurseries and cleaning prior to transplantation (Shafir et al., 2010).

Animal-assisted cleaning to control fouling organisms in coral culture has been performed while rearing coral juveniles settled from sexual reproduction. In ocean (in situ) nurseries, polyps of Acropora tenuis have been co-cultured with hatchery-supplied juveniles of the top shell Trochus niloticus (Omori, 2005; Omori, Iwao & Tamura, 2007). In laboratory (ex situ) nurseries, polyps of Pocillopora damicornis have been co-cultured with juveniles of the sea urchin Salmacis spaheroides and the gastropod Trochus maculatus collected from the wild (Toh et al., 2013). Both experiments have shown that the introduction of grazers under co-culture conditions controls algal proliferation, effectively increasing the survival of coral juveniles during their first 4 months of life. This period is critical in coral culture due to the vulnerability of juvenile corals, which may die due to smothering by filamentous algae. However, quantification of animal-assisted cleaning by coral reef fishes in the wild has not been tested until now.

In a large-scale coral reef restoration project in Seychelles (Indian Ocean), we deployed 9 mid-water rope nurseries, following methodology modified from Levy et al. (2010). Ropes loaded with farmed corals obtained through asexual propagation (fragments from donor colonies and rescued corals dislodged by storms or human activity) were floated 8 m below the sea surface to form a rope nursery. The entire structure was moored to angle bars hammered into the 17 m-deep sandy seabed. Each rope nursery held approximately 5,000 corals. The cleaning of biofouling organisms at each midwater rope nursery required 22.7 ± 9.78 (mean ± SD, n = 9) diver hours (range 12–36 diver hours) per month. Biofouling cleaning was repeated every 2–4 months. When we used a different type of midwater coral nursery (6 × 6 m PVC pipe frames layered with 5.5 cm-mesh tuna nets), reef fishes recruited at the nurseries consumed the biofouling organisms and reduced the required cleaning time (Frias-Torres et al., 2015). Net nurseries filled with corals provided a three-dimensional patch of habitat that facilitated the recruitment of resident fish. The lack of the three-dimensional habitat patch in the rope nurseries might explain why no resident fish community was recruited there, which made the periodic removal of biofouling organisms a necessity.

While cleaning the biofouling organisms at the midwater rope nurseries, we found that barnacles attached to the nursery ropes and to the coral/rope boundary were difficult to remove. After cementing nursery-raised corals to the reef restoration site, a mob of fish rammed the newly cemented corals to feed on mobile invertebrates that were recruited during the nursery phase or any leftover barnacles. The ramming fish species included the Sky Emperor (Lethrinus mahsena), Tripletail Wrasse (Cheilinus trilobatus), Titan Triggerfish (Balistoides viridescens) and flagtail triggerfish (Sufflamen chrysopterum; Table 1). Based on our dive logs, where we recorded the number of corals that were cemented and dislodged after each dive, coral dislodgement due to such fish attacks began when 13,140 corals were transplanted and increased to 16% of newly cemented corals when we reached 19,745 transplanted corals. This increase in coral dislodgement required repeating the cementing process towards the end of each dive, and hence, the total dive time required to complete our transplantation schedule increased (Fig. 1).

Figure 1 The problem.

As the number of transplanted corals increases, newly cemented corals are dislodged by hungry fish. The fish attempt to feed on barnacles and vagile invertebrates recruited to the corals during the nursery phase.

Table 1 Video-recorded barnacle and biofouling fish predators interacting with the experimental setup at the transplantation site.

Published trophic levels (mean ± SE) and diets are shown (Froese & Pauly, 2014; FishBase data, http://www.fishbase.org; Encyclopedia of Life, http://www.eol.org).

Scientific name	Common name	Fish base	
		Trophic level	Food items	
Barnacle predators	
Lethrinidae				
Lethrinus mahsena *	Sky Emperor	3.4 ± 0.42	Echinoderms, crustaceans, fishes, mollusks, tunicates, sponges, polychaetes and other worms.	
Labridae				
Cheilinus trilobatus	Tripletail Wrasse	3.5 ± 0.5	Mollusks, crustaceans, fish.	
Coris formosa	Queen Coris	3.3 ± 0.38	Mollusks, crustaceans, urchins.	
Balistidae				
Balistapus undulatus *	Orange-lined Triggerfish	3.4 ± 0.42	Benthic organisms: algae, echinoderms, fishes, mollusks, tunicates, sponges, hydrozoans.	
Balistoides viridescens	Titan Triggerfish	3.3 ± 0.44	Sea urchins, coral, crabs, other crustaceans, mollusks, tube worms.	
Sufflamen chrysopterum	Flagtail Triggerfish	3.5 ± 0.41	Wide variety of invertebrates.	
Bio-Fouling (excluding barnacles) predators	
Lutjanidae				
Lutjanus lemniscatus *	Yellowstreaked Snapper	4.0 ± 0.65	Benthic invertebrates	
Pomacanthidae				
Pomacanthus imperator	Emperor Angelfish	2.7 ± 0.00	Sponges, tunicates, other encrusting organisms	
P. semicirculatus	Semi-circle Angelfish	2.5 ± 0.0	Sponges, tunicates, algae	
Chaetodontidae				
Chaetodon auriga	Threadfin Butterflyfish	3.2 ± 0.5	Polychaetes, sea anemones, coral polyps, algae	
C. xanthocephalus *	Yellowhead Butterflyfish	3.0 ± 0.2	Benthic algae, coral polyps	
Pomacentridae				
Pomacentrus caeruleus	Blue-Yellow Damselfish	2.7 ± 0.30	Plankton, benthic algae, vagile benthic invertebrates	
Labridae				
Cirrhilabrus exquisitus	Exquisite Wrasse	3.4 ± 0.45	Zooplankton, vagile benthic invertebrate	
Coris aygula *	Clown Coris	3.4 ± 0.6	Hard-shelled invertebrates: crustaceans, mollusks, sea urchins	
Halichoeres nebulosus	Nebulous Wrasse	3.4 ± 0.5	Fish eggs, benthic invertebrates: crabs, sea urchins, ophiuroids, polychaetes, sponges, mollusks	
Labroides dimidiatus	Bluestreak Cleaner Wrasse	3.5 ± 0.5	Crustacean ectoparasites, fish mucus	
Scarus rubroviolaceus	Ember Parrotfish	2.0 ± 0.00	Benthic algae	
Pinguipedidae				
Parapercis hexophtalma	Speckled Sandperch	3.6 ± 0.3	Crabs, shrimps small fish	
Bleniidae				
Aspidontus dussumieri *	Slender Sabretooth Blenny	2.0 ± 0.00	Algae, detritus	
Acanthuridae				
Acanthurus leucocheilus *	Palelipped Surgeonfish	2.0 ± 0.0	Algae, detritus	
A. nigricauda	Blackstreak Surgeonfish	3.0 ± 0.40	Biofilm on sandy surfaces	
A. tennenti	Tennent’s Surgeonfish	2.0 ± 0.00	Benthic algae.	
Zebrasoma desjardini *	Sailfin Tang	2.0 ± 0.0	Filamentous algae, macroalgae, plankton	
Zanclidae				
Zanclus cornutus	Moorish Idol	2.5 ± 0.00	Sponges, sessile invertebrates	
Siganidae				
Siganus argenteus	Forktail Rabbitfish	2.0 ± 0.0	Algae	
S. stellatus *	Honeycomb Rabbitfish	2.7 ± 0.30	Benthic seaweeds	
Monacanthidae				
Cantherhines fronticinctus *	Spectacled Filefish	3.5 ± 0.37	Benthic organisms	
Ostraciidae				
Lactoria cornuta *	Longhorn Cowfish	3.5 ± 0.37	Benthic invertebrates	
Tetraodontidae				
Arothron meleagris *	Guineafowl Puffer	3.4 ± 0.6	Tips of branching corals, sponges, mollusks, bryozoans, tunicates, forams, algae, detritus	
Canthigaster valentini	Black Saddled Toby	2.8 ± 0.30	Filamentous green, brown and coralline red algae, tunicates, corals, bryozoans, polychaetes, echinoderms, mollusks	
Diodontidae				
Diodon hystrix *	Porcupinefish	3.4 ± 0.5	Hard shelled invertebrates: sea urchins, gastropods, hermit crabs	
D. liturosus *	Black-blotched Porcupinefish	3.4 ± 0.6	Crustaceans and mollusks.	
Notes.

* Species not recorded during visual underwater surveys of the transplantation site. Species are ordered taxonomically (Nelson, 2006).

Based on these field observations (the time invested in cleaning nurseries and coral dislodgement by fish), we searched for a biomimicry solution, i.e., a solution inspired by nature, to develop an innovative and sustainable technique (Benyus, 2002). Our inspiration was the cleaning stations at coral reefs where fish, sea turtles, sharks and rays congregate to be cleaned of parasites by cleaner fish and shrimps (Gorlick, Atkins & Losey, 1987; Losey, Balazs & Privitera, 1994; O’Shea, Kingsford & Seymour, 2010). Therefore, we systematically investigated the ability of coral reef fish to provide animal-assisted cleaning at coral nursery ropes prior to transplantation. Specifically, we hypothesized that (1) animal-assisted cleaning would occur closer to the bottom rather than higher in the water column, and (2) more animal-assisted cleaning would occur at the coral restoration site than at the nursery site because the periodic need to clean biofouling organisms at nurseries might result in a lack of an adequate biofouling-cleaning community at the nursery site. Here, we describe a novel technique for animal-assisted cleaning of nursery corals using a cleaning station. This new technique could be applied in coral reef restoration projects, particularly prior to the transplantation of corals on the reef and in the early nursery phase of sexually produced juvenile corals.

Materials and Methods

Field settings

This animal-assisted cleaning technique was developed as a result of two experiments conducted between 3 and 18 December 2013 at a coral reef restoration project within the marine protected area of Cousin Island Special Reserve, Seychelles, Indian Ocean (04°19′ 35″S; 055°39′24″E; Fig. 2).

Figure 2 Study area.

(A) Location of Cousin Island Special Reserve. (B) Detail of Cousin Island showing the nursery site and the rehabilitated reef (transplanted reef and control sites, healthy and degraded).

The restoration project included a nursery site and a reef transplantation site. The nursery site, located on the north-west side of the island at approximately 1 km from the nearest coral reef, included 9 mid-water rope nurseries. Each mid-water rope nursery consisted of 5 high-pressure PVC pipes (HP PVC), 600 × 64 mm in size, placed approximately 4 m apart, to which 20 m-long ropes were perpendicularly attached. Each rope held 80–150 corals, totaling approximately 5,000 corals in each rope nursery. The nurseries were attached to the 17 m-deep sandy seabed by anchor lines and maintained at a depth of 8 m below the sea surface by using recycled plastic jerrycans as buoys. The reef transplantation site, located on the south-west side of the island, consisted of a degraded coral reef affected by the mass coral bleaching event of the 1998, due to the coupling of the El Niño and the Indian Ocean Dipole (Spencer et al., 2000; Spalding & Jarvis, 2002) as well as the 2004 Indian Ocean Tsunami (Jackson et al., 2005). At this site, a gentle slope (roughly 25°) extends to a depth of 13 m. The seabed then flattens out and consists of a mixture of sand and coral rubble interspersed with granite outcroppings. The coral colonies grown in the midwater rope nurseries were transplanted to this degraded reef. At the time of the experiment, the transplantation site had been changed from a flattened-out degraded state to include 19,745 transplanted coral colonies of the following species: Acropora cytherea, A. damicornis, A. formosa, A. hyacinthus, A. abrotanoides, A. lamarki, A. vermiculata, Pocillopora damicornis, P. indiania and P. grandis.

Methods

A field permit was not required to conduct the experiments described herein at the marine reserve within the Cousin Island Special Reserve. The Special Reserve is managed by Nature Seychelles. As Nature Seychelles employees, we were able to perform underwater observations without the issuing of a specific permit at the no-take marine reserve, as long as we complied with the demands for no damage, harassment or taking of fish.

To test the animal-assisted cleaning of the nursery corals, we first assessed fish diversity at the transplantation site to determine whether the fish community could provide animal-assisted cleaning; then, we performed field experiments to quantify animal-assisted cleaning. To assess fish diversity at the transplantation site, we conducted visual underwater surveys via the standard point count method (Jennings, Boull & Polunin, 1996; Hill & Wilkinson, 2004; Ledlie et al., 2007). Briefly, divers were located at random points within the area, where they laid out a 7.5 m tape to form the radius of an imaginary cylinder and remained neutrally buoyant approximately 2 m off the seabed. All of the fish entering the 7.5 m cylinder radius were counted for 6 min and identified to the species level. During the seventh minute, each diver recorded cryptic fish species (hiding in the substrate) while swimming in a spiral from the center of the cylinder outwards. These point method counts were replicated six times.

To investigate animal-assisted cleaning, we developed an experimental unit resembling a ladder (Fig. 3 and Video S1). Angle bars were driven into the seabed 1.8 m apart, and mooring lines with buoys (recycled jerrycans filled with air and capped) were vertically attached to each angle bar. Ropes with nursery corals and biofouling organisms were then horizontally tied between the mooring lines like rungs on a ladder, at 0.3 m, 2 m, 4 m, 6 m and 8 m from the seabed. The experimental units were deployed at the nursery site (control) and the transplantation site (treatment) with 3 replicates per site. To avoid pseudo-replication, each experimental unit was placed at a different location within each site, and each replicate was arranged with a new set of coral nursery ropes. At the transplantation site, the experimental units were deployed on 9 December (2 locations) and 11 December 2013 (1 location). At the nursery site, the experimental units were deployed on 16 December (2 locations) and 18 December 2013 (1 location). The experimental rungs were cut from 20 m-long coral ropes from a midwater nursery filled with corals of opportunity (i.e., corals rescued from the seabed after breakage due to storms or anchor damage). The coral species included Acropora bruegemanni, A. formosa, A. abrotanoides, A. nobilis and A. robusta. The corals grew at the nursery for 17 months, and their size was 15.6 ± 4.51 cm (mean ± SD) at their maximum dimension (range, 11.5–21 cm). Therefore, the experimental units allowed the concept of a cleaning station to be tested by depth and site.

Figure 3 Experimental setup (testing the cleaning station) at both the nursery (control) and transplantation (treatment) sites.

(A) Schematic representation of the experimental setup showing the range of depths and elements. (B) Photograph of the setup with a diver. Credit for coral symbols: Woerner (2011). Photo credit: Casper van de Geer. See Video S1.

Small underwater cameras (GoPro) were placed next to the experimental units to remotely document the fish interacting with the nursery ropes without human disturbance. To analyze the fish assemblages at the transplantation site using underwater visual surveys and video recordings performed at the experimental units (both the transplantation and nursery sites), we generated an inventory of fish species and compared the video-recorded feeding behavior of each species to known trophic levels and food items reported in FishBase (http://www.fishbase.org) and the Encyclopedia of Life (http://www.eol.org). In FishBase, the mean trophic position is calculated based on all the food items consumed by a species, weighted by their relative abundance. The trophic level is obtained by adding 1 to the mean trophic position (Froese & Pauly, 2014). The range of trophic levels is as follows: 1 for primary producers, 2–2.19 for primary consumers (herbivores consuming mainly plants or detritus), 2.2–2.79 for omnivores (consuming plants or detritus and animals), 2.8–4 for secondary consumers and higher than 4 for tertiary consumers (carnivores).

To quantify animal-assisted cleaning, we counted the number of corals and barnacles on each of the nursery ropes prior to their placement at both sites (nursery and transplantation). The experimental units were then left in situ for 48 h, after which the number of barnacles was counted again, and the coral colonies were closely examined for signs of damage or predation. Barnacles were considered “eaten” when their calcareous exoskeletons were crushed and no barnacle soft tissue was present. Photographs of selected barnacle clumps and coral colonies were also taken before and after deployment for visual comparison.

Due to the variable number of barnacles per rope, the count data were converted into the percentage of barnacles eaten per rope. To meet the assumptions of parametric statistical analysis, the percentages were ArcSin transformed (Sokal & Rohlf, 1995). We tested the null hypothesis of no differences in the mean percentage of barnacles eaten per site and by depth using the transformed data in a two-way analysis of variance (ANOVA) model I (fixed factors). Post hoc comparisons were performed with Tukey’s Honestly Significant Difference (HSD) Test.

Results

The fish community at the transplantation site was diverse (51 species) and was dominated by wrasses (Labridae, 12 species; Fig. 4A). Other families present included surgeonfishes (Acanthuridae, 5 species), groupers (Serranidae, 4 species), damselfishes (Pomacentridae, 4 species), butterflyfishes (Chaetodonthidae, 4 species), triggerfishes (Balistidae, 3 species), goatfishes (Mullidae, 2 species), angelfishes (Pomacanthidae, 2 species) and dartfishes (Microdesmidae, 2 species). A total of 13 families were represented by only 1 species each. The species classified by trophic level included 11.8% herbivores, 15.7% omnivores, 50.9% secondary consumers and 21.6% carnivores. Based on these results, we determined that the transplantation site harbored a fish community capable of feeding on the biofouling organisms accumulated on the coral nursery ropes. Hence, we proceeded with the quantification of the animal-assisted biofouling cleaning using the experimental setup.

Figure 4 Fish assemblages.

Fish families, number of species per family, and trophic levels at the transplantation site during (A) underwater visual surveys (November 2013) and (B) video recordings of the experimental setup (December 2013). Insets in (A) and (B) show trophic groups (number of species per group indicated). Abreviations: Lab, Labridae; Aca, Acanthuridae; Ser, Serranidae; Pom, Pomacentridae; Cha, Chaetodontidae; Bal, Balistidae; Mul, Mullidae; Poc, Pomacanthidae; Mic, Microdesmidae; Sig, Siganidae; Tet, Tetraodontidae; Dio, Diodontidae; Oth, Other families with 1 species only, in (A) Lutjanidae, Bleniidae, Monacanthidae, Tetraodontidae, Carangidae, Apogonidae, Cirrhitidae, Syngnathidae, Lethrinidae, Pinguipedidae, Ephippidae, Synodontidae, Zanclidae and in (B) Lutjanidae, Bleniidae, Monacanthidae, Lethrinidae, Pinguipedidae, Pomacentridae, Ostraciidae, Zanclidae.

The field experiments revealed that the thornback boxfish (Lactoria fornasini, Ostraciidae) was the only species that was video recorded interacting with the experimental setup at the nursery site. This species feeds on benthic invertebrates and has a trophic level of 3.0 ± 0.0 (Froese & Pauly, 2014; FishBase data http://www.fishbase.org). Four additional species were video recorded at the nursery site but did not interact with the experimental setup: emperors (Lethrinus sp., Lethrinidae), goatfishes (Parupeneus sp., Mullidae), razorfishes (Xyrichtys sp. Labridae) and porcupinefishes (Diodon sp., Diodontidae). However, at the transplantation site, 32 fish species were observed feeding at the experimental setup (Figs. 4B, Fig. 5 and Table 1). Here, the video-recorded fish community (Fig. 5 and Video S1) was a subset of the species recorded during the visual underwater surveys and was also dominated by wrasses (Labridae, 7 species). Other families included surgeonfishes (Acanthuridae, 4 species), triggerfishes (Balistidae, 3 species), angelfishes (Pomacanthiade, 2 species), butterflyfishes (Chaetodontidae, 2 species), rabbitfishes (Siganidae, 2 species), pufferfishes (Tetraodontidae, 2 species), and porcupinefishes (Diodontidae, 2 species). A total of 8 families were represented only by 1 species each. The species classified by trophic level included 18.8% herbivores, 18.8% omnivores, 59.3% secondary consumers and 3.1% carnivores.

Figure 5 Animal-assisted biofouling cleaning.

(A) Barnacle predation at the transplantation site: the circle shows a clump of barnacles before (left) and 48 h after placement (right). (B) Titan Triggerfish, Balistoides viridescens, shown in the foreground of the experimental setup. (C) Reef fish lined up feeding on the 0.3 m coral rope at the transplantation site. Photo credit: Casper van de Geer. See Video S1.

The only species that was observed breaking and feeding on the barnacles attached to the ropes was the Titan Triggerfish (Balistoides viridescens; Fig. 5). Other biofouling predators were observed feeding on barnacle remains after the calcareous exoskeleton was broken by B. viridescens but otherwise fed on algae and sessile and mobile invertebrates attached to the ropes. No other animals were observed to interact with the experimental unit, although octopuses and sea turtles were recorded in the vicinity. The Humphead Parrotfish (Bolbometopon muricatum), a coral predator, was not video-recorded at the nursery or transplantation sites; however, it was a regular visitor feeding on biofouling fragments falling to the seabed when divers cleaned the nurseries. After the 48 h-long experiment, predation of corals (indicated by scarring or bite marks) was absent at both the transplantation and nursery sites.

Each 1.8 m-long rung in the experimental units harbored a similar number of corals: 13.4 ± 1.68 (mean ± SD) corals per rung at the transplantation site (range 10–16 corals) and 13.5 ± 1.84 corals per rung at the nursery site (range 10–17 corals). Despite these similar numbers of corals per rung, the average number of live and dead corals per rung varied. For instance, there were 7.9 ± 3.93 (mean ± SD) live corals per rung (range 2–14 corals) and 5.4 ± 4.22 dead corals per rung (range 1–14 corals) at the transplantation site, whereas there were 7.2 ± 3.43 live corals per rung (range 1–13 corals) and 6.2 ± 4.16 dead corals per rung (range 1–15 corals) at the nursery site. However, the differences were not significant (two-way ANOVA model I) between the transplantation and nursery sites (F1,56 = 2.88; p = 0.09) and between the number of dead and live corals per rung (F1,56 = 0.004; p = 0.94). Therefore, each independent replicate provided the same feeding substrate under the experimental field conditions.

Barnacle predation at the transplantation (treatment) site was 3.25 times higher overall than at the nursery (control) site (38.8% ± 0.21 SE and 12.2% ± 0.03 SE, respectively; F1,20 = 15.33, p = 0.0008). The depth of placement was critical. The highest barnacle predation was observed at the transplantation site on the 0.3 m ropes (94.8% ± 2.7 S.E.) and the 2 m ropes (83.3% ± 9.3 S.E.; F4,20 = 6.54, p = 0.002; Fig. 3). The site × depth interaction was significant (F4,20 = 10.16, p = 0.0001). Post hoc comparisons of the interaction term using Tukey’s HSD test revealed that barnacle predation at the transplantation site was similar on the 0.3 and 2 m ropes (p = 0.99), but it was 5–94 times higher compared with all other combinations of depths and sites (0.00029 < p < 0.003; Fig. 6).

Figure 6 Average barnacle predation.

Animal-assisted biofouling cleaning. Average barnacle predation per depth at the nursery (control) and transplantation (treatment) sites. Bars indicate standard error (n = 3).

Based on the results obtained from both experiments, we set up a dedicated cleaning station at the edge of the restoration site, away from the transplanted corals, marked by rebars hammered onto the hard substrate at 5 m intervals. We eliminated diver-assisted cleaning prior to coral transplantation; instead, we attached a nursery rope at the cleaning station, which was set at 0.3 m above the seabed, resembling the bottom rope shown in Figs. 3 and 5c. The cleaning station was located at the base of the mooring lines used by the divers to reach the transplantation site, and no significant increase in dive time was involved during its placement. After 48 h at the cleaning station, the ropes and coral/rope interface were free of barnacles and other biofouling organisms. The corals were also free of mobile invertebrates (except Trapezia sp. crabs). We used the cleaning station technique for 5 months (January–May 2014), and the rate of coral detachment due to fish attacks fell from an initial 16% to zero.

Discussion

Fish fed on barnacles and other biofouling organisms from the coral nursery ropes located at the transplantation site within 48 h of deployment of the experimental setup. The diverse fish community at the transplantation site, where 32 species from 4 trophic levels were recorded feeding at the experimental setup, ensured effective animal-assisted cleaning. The Titan Triggerfish (Balistoides viridescens) was the key species because it is capable of crushing the calcareous exoskeletons of the barnacles. The other fish species observed either fed on half-consumed barnacles left behind by B. viridescens or fed directly on the biofouling organisms found on the ropes and at the coral/rope interface. In contrast, the absence of a reef-associated fish community at the nursery site (only 1 species was observed feeding) explains the lack of animal-assisted cleaning of the experimental setup at this location.

At the transplantation site, animal-assisted cleaning occurred only on the coral nursery ropes located 0.3 and 2 m from the seabed. This depth suggests a safe zone or a maximum distance that reef fish will venture away from the protection of the seabed and reef structures to feed. Similarly, diurnal planktivorous reef fishes feed within a maximum distance from the reef; thus, they are able to dive safely back into the reef when exposed to predators (Hobson, 1993).

During the field experiments, the fish community video recorded at the transplantation site was a subset of the community observed through visual underwater surveys. Interestingly, 14 species were observed only at the experimental setup and were not recorded during the surveys (Table 1). The remote video recording performed to quantify fish feeding at the experimental setup while the divers were absent allowed us to register species that would otherwise have been hiding or that had fled during the visual underwater surveys. Such differences in census results obtained in the presence and absence of divers are consistent with previous quantifications of the diver effect on diver-based underwater visual fish censuses (Dickens et al., 2011).

Based on the video-recorded observations (Fig. 5 and Video S1) and known fish diets (Table 1), we deduced the relevant prey items that each trophic group consumed at the experimental setup, thus ensuring the animal-assisted cleaning of biofouling organisms. Herbivores consumed filamentous, coralline and benthic algae. Omnivores consumed vagile benthic invertebrates and sessile invertebrates (sponges, tunicates, other encrusting organisms). Secondary consumers consumed hydrozoans, bryozoans, sea anemones, echinoderms (sea urchins, ophiuroids), crustaceans (crabs, shrimps, barnacles), mollusks, tunicates, sponges, polychaetes and other worms as well as other vagile benthic invertebrates. Carnivores consumed benthic invertebrates. Because the video recordings were obtained during daylight hours, we were unable to quantify the biofouling predation of twilight and night predators, such as octopuses.

In summary, the reef fish at the coral transplantation site fed on biofouling organisms (algae, sessile and mobile invertebrates) on the coral nursery ropes prior to transplantation. The recorded barnacle predation rates were 95% and 83% on the ropes placed at 0.3 and 2 m from the seabed, respectively, 48 h after placement. The fish community at the transplantation site that provided the animal-assisted cleaning service included 32 species from 4 trophic levels: herbivores consumed filamentous, coralline and benthic algae; omnivores and secondary consumers consumed sessile and vagile benthic invertebrates; and carnivores consumed vagile benthic invertebrates. The animal-assisted cleaning technique derived from the experiments described herein consisted of a 48 h deployment of a coral nursery rope at the transplantation site within 0.3 m of the seabed (the “cleaning station”). Using the cleaning station, we reduced the dislodgement of newly cemented corals by fish from 16% to zero.

We suggest that future research could evaluate the incorporation of the cleaning station technique in two ways. First, the cleaning station could be included as a step prior to coral transplantation at geographic locations outside the Seychelles. The success of the technique will rely on the existence of key barnacle predators at each geographic location. Key barnacle predators also exist in other regions of the world. For example, in the western Atlantic ocean, stomach content analyses of the Hogfish (Lachnolaimus maximus), a large wrasse (Labridae), and the Batfish (Ogcocephalus nasutus, Ogcocephalidae) has revealed the presence of crushed barnacles (Randall, 1967). Therefore, investigating site-specific fish assemblages is essential to determine whether a cleaning station will be effective. Second, nurseries of sexually produced juvenile corals could become temporary cleaning stations. A key bottleneck in the mass culture of sexually produced corals is the high mortality of juveniles during their first 4 months post-settlement (Omori, 2005), due to their vulnerability to dying from smothering by filamentous algae (Toh et al., 2013). Therefore, during the first few months after coral settlement, in situ ocean coral nurseries could be placed within 0.3 m–2 m from the seabed at reef sites with an adequate grazer community. However, placing the coral nurseries permanently at a reef site until the corals reach transplantation size, to allow them to benefit from grazer activity, would increase the risk of damage to the nursery corals from coralivorous fish and anthropogenic impacts such as fishing and SCUBA diving (Levy et al., 2010). Therefore, once the juvenile corals have survived the early critical phase, the nurseries could be moved to a more permanent nursery site, away from the reef. Likewise, the nursery corals should be placed closer to the sea surface. Sexually produced coral juveniles would then benefit from the same rapid growth rates observed in the nurseries seeded via asexual coral propagation (Levy et al., 2010; Shafir & Rinkevich, 2010).

Here, we show that observations of animal behavior, biomimicry (i.e., obtaining inspiration from coral reef cleaning stations) and carefully designed experiments can provide innovative and sustainable solutions. We recommend such an approach to advance the emerging field of coral reef restoration.

Supplemental Information

Video S1 Experimental setup. Overview of setup and fish species feeding on sessile and mobile biofouling attached to the coral nursery ropes

This video provides an overview of the experimental setup at the transplantation (treatment) site and fish species feeding on sessile and mobile biofouling attached to the coral nursery ropes.

Click here for additional data file.

Supplemental Information 1 Raw data of baseline fish survey at transplantation site

This Excel file contains the raw data obtained during the baseline fish surveys conducted at the transplantation site, November 2013.

Click here for additional data file.

Supplemental Information 2 Raw data fish species feeding at experimental setup

This Excel file contains the raw data of species found feeding at the experimental setup at the transplantation (treatment) and nursery (control) sites, based on video recordings with GoPro cameras.

Click here for additional data file.

Supplemental Information 3 Raw data barnacle predation by site and depth

This Excel file contains raw data of the barnacle predation quantified at the experimental setup at the transplantation (treatment) and nursery (control) sites. Data are shown by site and depth.

Click here for additional data file.

We thank S Beach, M Beraud, S Clauson-Kaas, PH Montoya-Maya, C Reveret and K Rowe for their help during fieldwork; M Beraud for refinements of the experimental design and fish identification; C Reveret for his help in the experimental design; PH Montoya-Maya for the design of Fig. 2; K Henri and N Shah for managing the project in which this study was conducted; and PH Montoya-Maya, C Reveret and N Shah for manuscript comments.

Additional Information and Declarations

Competing Interests

Author Contributions

At the time of performing the study both SFT and CG were employees of Nature Seychelles in the Republic of Seychelles. During the writing and submission of the manuscript, SFT remains an employee of Nature Seychelles, while CG is an employee of the Local Ocean Trust in Kenya. From performing the study to manuscript submission, SFT is also a research collaborator with the Smithsonian Marine Station, FL, USA. The authors declare there are no competing interests.

Sarah Frias-Torres conceived and designed the experiments, analyzed the data, contributed reagents/materials/analysis tools, wrote the paper, prepared figures and/or tables, reviewed drafts of the paper, full draft write up and getting manuscript ready for publication.

Casper van de Geer conceived and designed the experiments, performed the experiments, analyzed the data, contributed reagents/materials/analysis tools, wrote the paper, prepared figures and/or tables, reviewed drafts of the paper.

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
