# Peer review of "Testing animal-assisted cleaning prior to transplantation in coral reef restoration"

_PeerJ, doi:10.7717/peerj.1287_

## Round 0.1 · original submission · Minor Revisions

The referees like your paper and make recommendaitons to improve it. Can you attend to these points and respond to them clearly to help me make a final decision. Could you also have an independent good English editor read over the paper to address the second referees suggestion? Please note that PeerJ does not proof read the English for you. Also, double check all figures will be legible when reduced in size in the publication (text on Fig 2?) because they will be published as submitted.

·

Basic reporting

No comments, see general comments.

Experimental design

No comments; see general comments.

Validity of the findings

No comments; see general comments.

Additional comments

Review: Testing animal-assisted cleaning prior to transplantation in
coral reef restoration

General comments:
This study is a simple and elegant solution to the problem that fish were dislodging corals from a transplantation site by feeding on biofouling fauna surrounding the nursery grown corals; by allowing the fish to clean the biofouling before transplantation. The solution to this problem was seemingly very simple and intuitive, but by demonstrating this in a clear experimental framework, this knowledge can be transferred to other projects and to the scientific community. Reef restoration efforts are rarely accompanied by such clear experimental design, hard data, and useful information. I am very impressed that this paper will set an example for future restoration work to solve problems not by guesswork, but by data driven experimentation. The paper has no experimental flaws and overall it is very well written. I have minor suggestions for making the tone a bit more scientific and slightly less colorful, but other than that I recommend publication with minimal delay.

Specific comments:
Introduction
Line 47: It should be made clear that the coral gardening concept does not necessarily require rope nurseries (that is just one possible approach).
Line 63: reporting the average and error without the count is not very informative: is this per nursery or per dive? What is the count?
Line 75-77; replace mob and rammed with more scientific terms.
Line 79; this seems like an unusually precise number; is this an approximation?
Line 81; suggest replacing forced us with required repeating…
Line 85; suggest delete “hungry”, the fish could have been ‘full’ and just mean…the assumption is probably correct, but it’s not necessary to make.
line 106: what was the diameter of the PVC?
Results:
Line 194: replaced we were certain, with we determined.

Line 207 -212; suggest referencing the figure.
Line 309: typ-o 316?

Line 315; suggest introducing the term cleaning station in the methods and sticking with that term in the figures as well.
Line 318: Last line of the discussion is one of the most important lines in the paper; what closing message do you want to leave with the reader. I think you might mention in the intro that restoration projects are rarely very data or experimentally driven and that this study is an example how relatively straight forward experiments can provide clear solutions to problems. You should also add somewhere in the discussion recommendations for future work. What about the interactions between access by the fish community and coral growth?

·

Basic reporting

This is a useful study and the results warrant publication. However, while the manuscript is intelligible, the grammar needs a little work in places. I would suggest this manuscript would benefit from a thorough edit to improve continuity and remove a few pieces of redundant text. For example I would elite the sentence beginning on l68 "Such support is absent..."

I think the ms would benefit also from some effort to simplify and strive for brevity. The essential message is that following observations that biofouling associated with nursery reared corals attracted fish which increase the risk of post-transplantation dislodgement. Hence the study explored a novel approach to biological management of fouling organisms etc etc

Experimental design

Straightforward design principles were used, including adequate levels of replication.

I feel that there should be more detail provided on the assessment of the post-transplantation dislodgment. The problem as outlined in Figure 1 is apparently resolved after the cleaning station treatment approach is used, but it is difficult to understand how the dislodgment data was gathered in each case. For example, what was the density of transplantation and when or how frequently post-transplantation was the level of detachment measured.

Validity of the findings

The results seem sound and the information of use. Biological control of fouling organisms has been undertaken elsewhere, including with transplanted corals, eg. Omori, 2005,2007, so I would encourage the authors to broaden their exploration of the literature around this topic to place their work into a broader context.

Omori M (2005) Success of mass culture of Acropora corals from egg
to colony in open water. Coral Reefs 24:563

M. Omori, K. Iwao, M. Tamura (2007) Growth of transplanted Acropora tenuis 2 years after egg culture. Coral Reefs 27(1): 165

The authors would likely strengthen the significance of this work if they could assess the cost-benefit of the staged cleaning area approach relative to other protocols for nursery culture and transplantation.

Similarly, I feel that the observations on the net-based nursery being more effective that just a rope system unless the rope system is established over a cleaning area supporting diverse fish community should be elaborated on further, as to the pros and cons or relative costs of both culture systems.

Additional comments

I think this work is innovative and suggests a few new lines of research worth exploring in the development of more cost effective coral culture. The authors may wish to explore the effect of nursery culture over such cleaning station habitats on smaller coral colonies, as a key bottleneck to coral production remains the high mortality in the first year post-settlement when sexually produced juveniles are cultured.. It would be useful therefore to know if culture from the very smallest sizes of viable fragments in this type of situation would bring improvements

---

## Round 0.2 · accepted · Accept

Thank you for the thorough and clear revisions.

---

## Author Rebuttal · Round 0.2

Dr. Sarah Frias-Torres
Nature Seychelles, Amitie, Praslin, Seychelles
And
Smithsonian Marine Station,
1420 Seaway Drive, Fort Pierce, FL 34949, USA
*sfriastorres@gmail.com*
Tel (USA)☺+1)  772-462-6220
Tel (Seychelles): (+ 248) 278-08-11

9 September 2015

Attn: Dr. Mark Costello
Academic Editor
On-Line Submission

Dear Dr. Costello,

This is a revision of our manuscript **# 2015:07:5638:0:1:REVIEW**  titled:

"Testing animal-assisted cleaning prior to transplantation in coral reef restoration"

by S. Frias-Torres & C. van de Geer

We are grateful for the time you and the reviewers spent evaluating the manuscript. We hope this revised manuscript will comply with the required minor revisions suggested and be accepted for publication at *PeerJ*.

Since our first submission, we had an article accepted for publication:

Frias-Torres S, Goehlich H, Reveret C, Montoya-Maya PH. 2015. Reef fishes recruited at mid-water coral nurseries consume biofouling and reduce cleaning time in Seychelles, Indian Ocean. *African Journal of Marine Science*.

This article is in our reference list. We have been assigned a doi (10.2989/1814232X.2015.1078259) but we have not received the proofs yet so the article is not posted at the journal website. We consider this article published.

As per your suggestion, after adding all the changes requested by the reviewers, we sent the manuscript for review by two independent science editors at the NPG (Nature Publishing Group) Editing Services. The resulting professionally edited manuscript is the version we provide as the final document.

The table below shows every comment or request provided by the reviewers and the action we have taken. Due to the editing and new text additions, the line numbers indicated by the reviewers do not necessarily match the line number in the revision. As per your request, changes in the "Tracked Changes" version of our manuscript (Word file) are shown in yellow highlight.

.

| Comment/Correction | Action |
|---|---|
| ACADEMIC EDITOR: Dr. Mark Costello | |
| Have an independent good English editor read over the paper to address the second referees suggestion | Manuscript edited by two independent English editors at NPG Editing Services |
| Double check all figures will be legible when reduced in size in the publication (text on Fig 2?) | All figures legible |
| REVIEWER 1: Dr. Zac Forsman | |
| Line 47: It should be made clear that the coral gardening concept does not necessarily require rope nurseries (that is just one possible approach). | Corrected |
| Line 63: reporting the average and error without the count is not very informative: is this per nursery or per dive? What is the count? | Corrected |
| Line 75-77; replace mob and rammed with more scientific terms. | We want to keep both words to help visualize readers what we observed. Here, we use "mob" with the meaning of "large group of [people or other animals] out of control", and "rammed" as "strike with a heavy impact, as a battering ram". We can't find more scientific terms to provide readers with a visual reference. |
| Line 79; this seems like an unusually precise number; is this an approximation? | This is the exact number of corals transplanted to that point. Every day, we registered in our dive logs how many corals had been transplanted, so we knew precisely the total number of corals transplanted at any given quarter of the year. |
| Line 81; suggest replacing forced us with required repeating… | Corrected |
| Line 85; suggest delete "hungry", the fish could have been 'full' and just mean…the assumption is probably correct, but it's not necessary to make. | Deleted |
| line 106: what was the diameter of the PVC? | Diameter and PVC type added |
| Line 194: replaced we were certain, with we determined. | Corrected |
| Line 207 - 212; suggest referencing the figure | Figure referenced |
| Line 309: typo 316? | Corrected |

| Comment/Correction | Action |
|---|---|
| Line 315; suggest introducing the term cleaning station in the methods and sticking with that term in the figures as well. | Cleaning station term shown now in abstract, keywords, introduction, methods, and Fig 3 legend |
| Line 318: Last line of the discussion is one of the most important lines in the paper; what closing message do you want to leave with the reader. I think you might mention in the intro that restoration projects are rarely very data or experimentally driven and that this study is an example how relatively straight forward experiments can provide clear solutions to problems. You should also add somewhere in the discussion recommendations for future work. What about the interactions between access by the fish community and coral growth? | We agree some restoration projects lack proper experimental design, but issuing a generalized statement such as the one proposed here will require the back up of a comprehensive literature review of every coral restoration project started to date. Such a comprehensive review is beyond the scope of this paper. Focusing on the last line, we have added recommendations for future work, including access by the fish community to corals in the nursery during their growth phase. |
| REVIEWER 2: Dr. Andrew Heyward | |
| This manuscript would benefit from a thorough edit to improve continuity and remove a few pieces of redundant text | Manuscript edited by two independent English editors at NPG Editing Services |
| Delete the sentence beginning on L68 "Such support is absent..." | Corrected |
| I think the ms would benefit also from some effort to simplify and strive for brevity. The essential message is that following observations that biofouling associated with nursery reared corals attracted fish which increase the risk of posttransplantation dislodgement. Hence the study explored a novel approach to biological management of fouling organisms etc etc | We strive for brevity while adding the new information requested by reviewers |
| I feel that there should be more detail provided on the assessment of the posttransplantation dislodgment. The problem as outlined in Figure 1 is apparently resolved after the cleaning station treatment approach is used, but it is difficult to understand how the dislodgment data was gathered in each case. For example, what was the density of transplantation and when or how frequently posttransplantation was the level of detachment measured. | We explain how we recorded cementing and dislodgement in our dive logs |
| Biological control of fouling organisms has been undertaken elsewhere, including with transplanted corals, eg. Omori, 2005,2007, so I would encourage the authors to broaden their exploration of the literature around this topic to place their work into a broader context | We added the information of co-culture with grazing animals in the Introduction |
| The authors would likely strengthen the significance of this work if they could assess the cost benefit of the staged cleaning area approach relative to other protocols for nursery culture and transplantation. | We agree a detailed cost-benefit analysis comparing our cleaning station technique with other protocols for nursery culture and transplantation is needed. However, we think such detailed analysis is beyond the scope of this paper. We hope to address the issue in a different paper. |

| Comment/Correction | Action |
|---|---|
| I feel that the observations on the net based nursery being more effective that just a rope system unless the rope system is established over a cleaning area supporting diverse fish community should be elaborated on further, as to the pros and cons or relative costs of both culture systems. | We have now addressed this comment towards the end of the Discussion section. |
| The authors may wish to explore the effect of nursery culture over such cleaning station habitats on smaller coral colonies, as a key bottleneck to coral production remains the high mortality in the first year post-settlement when sexually produced juveniles are cultured. It would be useful therefore to know if culture from the very smallest sizes of viable fragments in this type of situation would bring improvements | We have now addressed this comment towards the end of the Discussion section. |

We look forward to hearing from you soon.

Sincerely
Sarah Frias-Torres, Ph.D.
Chief Scientist & Coordinator, Reef Rescuers, Nature Seychelles, Republic of Seychelles
Research Collaborator, Smithsonian Marine Station, Fort Pierce, FL, USA

cc / Casper van de Geer [cvandegeer@gmail.com]